# Safety climate in the operating room in the pre-pandemic and pandemic period of COVID-19: A mixed method study

**Rosilene Alves Ferreira**[1‡]*, **Cintia Silva Fassarella**[1‡], **Olga Maria Pimenta Lopes Ribeiro**[2], **Rosane Barreto Cardoso**[3], **Danielle de Mendonça Henrique**[1], **Flávia Giron Camerini**[1], **Rogério Marques de Souza**[4], **Ricardo de Oliveira Meneses**[1]

1 Department of Medical-Surgical Nursing, School of Nursing, State University of Rio de Janeiro, Rio de Janeiro, Brazil, 2 Nursing School of Porto, Higher School of Nursing of Porto, Porto, Portugal, 3 Department of Fundamental Nursing/History of Nursing, Anna Nery School of Nursing, Federal University of Rio de Janeiro, Rio de Janeiro, Brazil, 4 Head of the Nursing Unit of the Surgical Center, University Hospital Pedro Ernesto, Rio de Janeiro, Brazil

☯ These authors contributed equally to this work.
‡ RAF and CSF co-first authorship on this work.
* rosilene.alvesferreira.uerj@gmail.com

## Abstract

### Introduction

To verify whether the COVID-19 pandemic has had an impact on the safety climate based on the perception of the multiprofessional team in the operating room and to analyze the domains of the safety climate during the pre-pandemic and pandemic period of COVID-19, demonstrating the intersections of quantitative and qualitative approaches.

### Methods

Mixed-method research using a convergent approach strategy, carried out in the operating room of a university hospital, located in Rio de Janeiro, Brazil. The nature of the quantitative phase was cross-sectional, and the nature of the qualitative phase was descriptive. We used the Pillar Integration Process to integrate the data. This research considered the pre-pandemic period was defined as before March 2020 and for the pandemic period, the 2nd and 3rd global waves. Research was approved by the institution's board management and ethics committee.

### Results

145 health professionals participated in the quantitative approach, and 20 in the qualitative approach. The impact of the COVID-19 pandemic was highlighted in the domains 'Perceived stress' (p-value = 0.017); 'Working conditions' (p-value = 0.040). Six categories emerged from the qualitative analysis, namely: Stress and professional performance due to COVID-19; Patient safety protocols in the operating room; Responsibility for patient safety, lack of effective communication and performance feedback; Biosafety of the professional staff in the operating room; Security culture maturity; Fair culture, organizational learning, and reporting mistakes. As a result of the data integration, 6 pillars were identified:

**Data Availability Statement:** All files are available from the database Zenodo (doi.org/10.5281/zenodo.10892777).

**Funding:** This article was supported by the Coordination for the Improvement of Higher Education Personnel - Brazil (CAPES) - Funding Code 001, awarded to RAF and CSF. The funders had no role in study design, data collection and analysis, decision to publish, or preparation of the manuscript.

**Competing interests:** The authors have declared that no competing interests exist.

Perception of communication in the operating room; Evolution of safety culture; Overview of protocol management and implementation; Fair organizational culture; Perception of stress due to COVID-19; Perception of professional performance due to COVID-19.

## Conclusions

The impact that COVID-19 had on the safety climate in the operating room is evident. It underlines the need to implement strategies that support the solidification of attitudes aimed at patient safety, even in emergencies.

## Introduction

A safety climate comprises the healthcare professional's perception of how the organization manages patient safety issues. It is all about the measurable part of patient safety culture, and we can define it as the product of values, attitudes and behaviors of groups and individuals that demonstrate commitment to health and safety management. The perception that the professionals of an organization have in relation to different fields of safety climate is an important tool for analyzing the safety culture of the healthcare institution [1–3].

In recent decades, the increased complexity of surgical procedures and health care, following technological and scientific developments, makes patient safety promotion one of the main challenges in the different contexts of health systems, including the operating room environment, as it is one of the hospitals' sectors with the highest incidence of adverse events [4–6]. Thus, by researching the safety climate in the operating room, it is possible to identify negative and positive points in a highly specialized setting with high incident rates. Managers can use safety climate assessment tools for periodic benchmarking purposes, for prospective trend analysis based on the implementation of improvement actions and during critical times, as in the case of the COVID-19 pandemic [7].

Daily, the number of hospitalizations due to COVID-19 increased, thus it requires short-term organizational planning to improve professional practice environments, guaranteeing health care quality and safety conditions, in the face of a context of rapid change and uncertainty [8, 9]. Within this scope, discussion on safety climate in the context of COVID-19 becomes imperative considering that the pandemic has gradually impacted the surgical practice environment and the health organizations' routines [5, 10].

It is essential to ponder over the impact of the pandemic on the safety climate and, in this sense, there is evidence of the consequences such as the postponement of surgeries, an increase in clinical complications due to the absence of surgical intervention; the need to implement new biosafety protocols and modifying workflows [11]. In addition, growing concerns during the COVID-19 pandemic have been related to increased workload, increased complexity of health care, stress, time pressure and limited resources, postponement of surgeries, increased complications due to the lack or postponement of surgical interventions, risk of transmission of COVID-19 during procedures, awareness of the rational use of personnel, and collective protective equipment. Hence, it has had a negative impact on some of the safety climate fields, with repercussions on the quality of health care and the safety of surgical patients [5, 10, 12, 13].

Worldwide, enormous challenges arise from the COVID-19 pandemic, and there is evidence that the pandemic has potentiated existing failures and latent conditions in the health system, however the impact on these variables needs further research [5, 14]. Given the encounters with adversity caused by the COVID-19 pandemic, it is estimated that having

knowledge of and evaluating the safety climate of the perioperative environment makes it possible to establish a situational diagnosis enabling education, implementation and monitoring actions, combining strategies to improve work processes, contributing to quality, safety for patients and, surgical staff [7, 13, 15, 16].

### Aim

We chose the following research problem: has the COVID-19 pandemic had an impact on the safety climate in the operating room? The objectives of this study were to verify whether the COVID-19 pandemic had an impact on the safety climate from the perception of the multiprofessional team in the operating room and to analyze the domains of the safety climate in the pre-pandemic and pandemic period of COVID-19 from the perception of the multiprofessional team in the operating room, highlighting the intersections of quantitative and qualitative approaches.

## Methods

### Design

We used mixed-method research and applied a convergent approach strategy that made it possible to identify convergences and divergences between the data. The quantitative data was cross-sectional, and the qualitative data was descriptive.

Convergent approach strategy collects quantitative and qualitative data concurrently and analyzes it separately, following we proceed with the integration of the separate results. First, inferences can be made from the findings of each method and subsequently these inferences can be assembled in the discussion for a meta-inference; second, the findings of each method can be gathered in the discussion using tools for data integration. This may happen if the aim is to identify convergence, divergence, and discrepancy, in which different methods are used to study the same question [17].

Given the complexity of the safety climate study, and the fact that the operating room is so affected by an atypical context such as the COVID-19 pandemic, the integration of quantitative and qualitative data permits analysis and conclusions to provide a complete view of the object under study, since the complexity of the research problem requires answers that go beyond statistical analysis or researching the meaning of a phenomenon. The mixed method requires in-depth analysis and a combination of the two approaches to obtain a more complete and in-depth analysis of the research problem [17–20].

To conduct the study and ensure greater reliability, we used the EQUATOR network tools [21], guaranteeing methodological rigor, quality, and transparency in scientific writing. We applied the Strengthening the Reporting of Observational Studies in Epidemiology (STROBE®) [22] tool for the cross-sectional quantitative phase, the Consolidated Criteria for Reporting Qualitative Research (COREQ) [23] for the qualitative phase and the Mixed Methods Appraisal Tool (MMAT) [19, 24] for the integration of quantitative and qualitative approaches.

### Scenario study

The study was carried out in the operating room of a university hospital located in Rio de Janeiro, Brazil, which opened its doors in the 1950s. It is a large hospital with 525 beds and more than 60 specialties and subspecialties. The facility has sophisticated technology for cardiac surgery, heart transplantation, gender reassignment, urological, gynecological, plastic, general, and kidney transplantation, neurosurgery, pediatrics, orthopedics, and vascular and thoracic surgery. In addition to outpatient referral services in various health fields with multiprofessional care.

At the beginning of 2019, the da Vinci Xi robotic device, was launched at the service, and consequently it enabled robotic surgeries in the urology, gynecology, thoracic, head and neck specialties. The operating center has 20 operating rooms, one of which is for internal emergencies and the others for elective surgeries, a hybrid room used exclusively for vascular surgery, a room exclusively used for heart surgery and heart transplantation, a room for robotic surgery and a post-anesthetic recovery room, which has the capacity to see 12 patients simultaneously.

## Population and sampling

The study population consisted of all the professionals in the study setting (N = 224 professionals). In the quantitative phase, we calculated the sample size as a simple random sample with a 95% confidence level using the Epi Info v5.5.9 software tool, resulting in a sample of 141 professionals.

For the qualitative phase, we chose purposive sampling, hence it was possible to deliberately select individuals who could provide the required information and who had experienced the central phenomenon or key concept under study [20]. We selected the participants who could best contribute to the problem and the research question, thus ensuring the participation of all health professional categories, who worked in the operating room in the pre-pandemic and pandemic period of COVID-19.

We used the following inclusion criteria: health professionals working in the operating room with direct or indirect interaction with patients; who worked during the pre-pandemic and pandemic periods of COVID-19; professionals with a minimum workload of 20 hours/ week and followed the guidelines of the instrument used for data collection.

For the qualitative phase, the inclusion criteria were anesthesiologists, surgeons, and nurses, as well as nursing and support technicians, who worked during the pre-pandemic and pandemic periods of COVID-19; with a minimum workload of 20 hours/week; having participated in the quantitative phase of the study; and played a leadership role in the operating room.

We highlight the fact that in the present study, the pre-pandemic period of COVID was defined as March 2020, since the World Health Organization on March 11[th], 2020, declared the novel coronavirus (COVID-19) outbreak a global pandemic. As for the pandemic period, this research considered the 2[nd] and 3[rd] global waves (November 2020 to December 2021 and December 2021 to May 2022, respectively) [25, 26].

## Data collection instruments

In what concerns the quantitative study, we used the validated self-administered Safety Attitudes Questionnaire/Operating Room (SAQ/OR), as it is a version that has been translated, adapted, and validated for the Brazilian context and because it is specific to the study scenario, as it is a specific instrument for assessing the safety climate in the operating room [7]. It contains 40 items, divided into six domains, namely: safety climate; perception of management; perception of stress; working conditions; communication in the surgical environment; and perception of work performance [7].

We scored the answers as 'Strongly disagree' - 0 points; 'Partially disagree' - 25 points; 'Neutral' - 50 points; 'Partially agree' - 75 points; 'Strongly agree' - 100 points; and 'Not applicable', which did not generate a score. We grouped the subsequent scores by domain, and we gave a score to each domain by calculating the average of their sum. Scores can vary from 0 to 100, with values of 75 or more representing a positive perception of the patient safety climate [7].

We collected the qualitative data using a semi-structured interview script containing six open questions based on the domains of the SAQ/OR instrument: 'How do you rate the safety

climate in the surgical environment before and during the COVID-19 pandemic?'; 'How do you identify management actions before and during the COVID-19 pandemic? Please, exemplify'; 'With regard to stress, how do you feel on a daily basis when carrying out your activities in the operating room during the COVID-19 pandemic and how did you feel before the pandemic?': 'How do you rate your working conditions before and during the COVID-19 pandemic?'; 'Regarding communication in the surgical environment, how do you evaluate the transmission of information and the information tools used during the COVID-19 pandemic and before the pandemic?'; and 'With regard to professional performance, how do you consider and evaluate your work before and during the COVID-19 pandemic?

### Data collection

We collected, quantitative data, from August 28, 2021, to July 29, 2022, and we gathered, qualitative data, from June 8th to July 29th, 2022, meeting the study's inclusion criteria. We chose this specific period because of the pandemic. Thus, the study scenario respected the epidemiological criteria of the region with the return of elective surgeries as well as the presence of students and researchers in the unit. Hence, a longer collection period was necessary to reach the desired sample. Above all, it guaranteed the safety of both professionals and researchers due to the restriction of the flow of the professionals in the unit.

For the quantitative phase, the professionals in the study setting who met the inclusion criteria for this research, were invited personally and physically to participate in the research, guaranteeing health care during the COVID-19 pandemic. After explaining the purpose of the study, and the need to express their agreement to participate in the study by signing the Free and Informed Consent Form, we gave them a printed data collection instrument to fill in, which took approximately 10 minutes. These were collected, stored, and coded according to their feedback.

In the qualitative phase, the main researcher of this study conducted the interviews. The researcher established a relationship by approaching the participants individually and face-to-face in the study setting, ensuring multi-professional participation. After the researcher's presentation, explaining the purpose of the interview and of the study, we analyzed the data and invited them to participate in the voice recording and to express their acceptance by filling in the Term of Authorization for the Use of Testimony.

It is worth noting that we carried out pilot interviews to ensure that the questions placed would not induce the participants' responses and that they would achieve the study's objective, and we assured that they were not a part of the data analysis. After adjusting the necessary questions placed in the pilot interviews, we began data collection by conducting interviews, which were audio recorded on a digital device with flash drive support and were then coded according to the sequence of the interview and professional category, lasting approximately 15 to 30 minutes.

The environment chosen for the interview was the support room, and we did not allow any interruptions to let the participants feel free to express their individual ideas, feelings, needs and opinions. Considering the pandemic, we respected 1.5 meters between researcher and participant, and both wore facemasks throughout the entire interview.

### Data processing and analysis

We manually entered the quantitative data into a Microsoft Excel® spreadsheet, and then entered it into the R statistical software, version 4.2.1, a freeware tool used for analyzing and processing statistical data. We used descriptive statistics to analyze the numerical and categorical variables. For the description of the team's profile, we presented the categorical variables

(gender, ethnicity, category, and team) in absolute and relative frequencies. To analyze the differences in the perception of professionals in relation to the domains of the safety climate, during the pre-pandemic and pandemic periods of COVID-19, we used the mean standard deviation and chi-square test ($X^2$). The significance level used was 0.05.

We used the software Interface de R pour les Analyses Multidimensionnelles de Textes et de Questionnaires (IRaMuTeQ) to analyze the qualitative data. To organize the qualitative data, the main author, transcribed all the interviews in full and he later revised them to confirm that he had not omitted any words. The main author also used the LibreOffice software and identified each interview with a number according to the order that he carried out the interviews. This procedure guaranteed the anonymity of the participants, and it only used the job title as identifiers, namely: Nurse; Nursing technician; PACU nursing technician; Surgeon and Anesthesiologist. Following, we subjected the data to discursive textual analysis and interpreted it according to the thematic categories generated by the descending hierarchical classification—CHD [27].

To integrate the quantitative and qualitative data in a transparent and rigorous manner, we followed the steps of the Pillar Integration Process (PIP) model for data integration in mixed methods research. PIP is a joint display technique for integrating quantitative and qualitative data that have undergone separate initial analysis for the same case and we are able to study it together. It also aims to minimize observer bias and maximize opportunities for synthesis, both visually and methodologically [28].

In what concerns the convergent approach technique, we separately analyzed the quantitative and qualitative data, and then integrated both results by using the PIP so that convergences and divergences could be underlined in the data intersection section. PIP comprises four phases, namely: listing, matching, checking, and pillar building. Each phase is completed sequentially after the initial quantitative and qualitative analyses have separately been completed [28]. Phase 1, listing, consisted of listing the raw data and selecting quotes. This data is listed in the joint display in the QUANT Data and QUAL Codes columns, respectively.

In the second phase, we matched the data listed in the QUANT and QUAL columns. We continued this matching process by combining the quantitative data in the QUANT Categories column with the qualitative data in the QUAL Categories column. This phase involved reflecting on the content related to the initial data listed, aligning similar data horizontally, refining and organizing the categories.

Each list was organized and compared on the joint display lines so that the qualitative items reflect parallel patterns, similarities, or any other relational quality with the quantitative items. At the end of phase 2, the QUANT Data, QUANT Categories, QUAL Codes and QUAL Categories columns were filled in. We checked all data in the four filled outer columns for completeness to ensure that the rows were properly matched.

The checking procedure took place after combining the data to confirm the matching accuracy. All the data in the four filled outer columns were checked for completeness to ensure that the rows were properly matched. To ensure that no raw data was lost, we carried out the checking procedure twice.

In the last phase—pillar building—the pillar was built in the central column after comparing the findings that developed in the previous phases. We conceptualized insights and as a result new categories emerged, they were identified by connecting and integrating the quantitative and qualitative columns, by establishing inferences about which patterns, perceptions or themes emerged and possible explanations. We described the categories resulting from the integration of quantitative and qualitative data in the central PILLAR column of the joint display.

## Ethics

Research was approved by the institution's board management and by the coordinator of the operating room. It was also submitted to and approved by the research ethics committee. The professionals who agreed to participate in the study signed a Free and Informed Consent Form (FICF) and an authorization form for the use of testimony. The Ethics Committee approved this research under substantiating opinion n°. 3.138.243 and an amendment approved under substantiating opinion n°. 4.638.445.

# Results

## Quantitative data

In the quantitative phase, 145 health professionals participated in the study, with a prevalence of female employees (54%). Regarding the professional group, nursing staff predominated, and as for the professional category, nursing technicians (circulator/instrumentalists) were the majority, as shown in Table 1.

The quantitative data demonstrates that only the 'Communication in the surgical environment' domain can be considered positive in both periods. This result is relevant, as this domain is the differentiating factor of the SAQ/OR to SAQ. In addition, the COVID-19 pandemic has had an impact on the safety climate, specifically in the 'Perception of stress' and 'Working conditions' domains, showing a statistically significant association, with a p-value of less than 0.05, with p-values of 0.017 and 0.040, respectively (see Table 2).

## Qualitative data

The study sample consisted of 20 professionals: 03 (15%) anesthesiologists, 02 (10%) surgeons, 05 (25%) nurses, 09 (45%) nursing technicians and 01 (5%) support professional. The study sought to ensure a balance between the professional categories, by a greater representation of nursing technicians. The predominant gender was female (65%). The participants interviewed

**Table 1. Characteristics of health professionals in the operating room.**

| Variáveis | Categories | N | % |
|---|---|---|---|
| Sex | Female | 79 | 54 |
| | Male | 66 | 46 |
| Ethnicity | White | 68 | 47 |
| | Brown | 35 | 24 |
| | Black | 41 | 28 |
| | Indigenous | 1 | 1 |
| Job position | Nursing technician | 46 | 32 |
| | Surgical technician/Circulating nurse | 23 | 16 |
| | Surgery | 18 | 13 |
| | Nurse | 17 | 12 |
| | Surgery Resident | 16 | 11 |
| | Support team | 15 | 10 |
| | Anaesthesiologist | 5 | 3 |
| | Anaesthesiologist Resident | 5 | 3 |
| Team | Nursing | 86 | 60 |
| | Medical | 44 | 30 |
| | Support | 15 | 10 |

**Table 2. Association of safety climate domains by pre-pandemic and pandemic periods of COVID-19.**

| Domains | Pre-pandemic | | Pandemic | | X² | p-value |
|---|---|---|---|---|---|---|
| | mean | SD | mean | SD | | |
| Safety Climate | 69.75 | 20.24 | 70.21 | 20.49 | 0.0025 | 0.960 |
| Perception of Management | 65.68 | 21.17 | 65.13 | 21.84 | 0.0769 | 0.782 |
| Perception of Stress | 54.87 | 27.31 | 54.17 | 27.10 | 5.6709 | **0.017** |
| Working Conditions | 62.44 | 21.21 | 63.30 | 21.23 | 4.2237 | **0.040** |
| Communication in the Surgical Environment | 75.52 | 17.67 | 76.82 | 17.61 | 0.2907 | 0.590 |
| Perception of Job Performance | 29.48 | 24.85 | 28.66 | 24.88 | 1.1679 | 0.280 |

are statutory employees, who play leadership roles in the operating room, management of the operating room and coordination of anesthesiology and surgical charting.

The general corpus consisted of 20 texts, separated into 494 text segments (ST), and 409 ST were used (82.79%). 17,409 Occurrences (words, forms or words) emerged, of which 2,256 were distinct words and 1,181 had a single occurrence. In their statements, professionals described the level of stress they went through while carrying out their duties in the operating room in the pre-pandemic and pandemic periods of COVID-19.

They talked about the protocols carried out in the operating room and the attitudes towards complying with the protocols, comparing them before and during the pandemic, to guarantee patient safety. The statements expressed the responsibility for patient safety, the lack of communication and feedback on professional performance and, revealed the professionals' assessment of the management's concern to establish protective measures and the availability of the necessary materials, and how it was dissolving as the peaks of the pandemic were overcome.

Initially, they pointed out the service's weaknesses and the difficulty in understanding that safety in the surgical environment involves a set of values and attitudes. According to the statements, the participants feature that it was essential to improve the maturity of the organizational culture to guarantee an effective safety climate, by constantly introducing improvement strategies, as well as the existence of a quality-of-service sector. The statements also expressed the position in relation to the occurrence of errors and the difficulties concerning communication and accountability for mistakes, indicating the lack of a fair culture and organizational learning. The professionals' statements reflect the impact of the pandemic on the patient safety climate in the operating room.

The participants' statements were interpreted according to the thematic categories caused by the CHD and classified into six different categories, namely: Category 1, 'Stress and professional performance due to COVID-19', with 73 TS (17.85%); Category 2, 'Patient safety protocols in the operating room', with 49 TS (11.98%); Category 3, 'Responsibility for patient safety, lack of effective communication and feedback on performance', with 58 TS (14.18%); Category 4, 'Biosafety of the professional staff in the operating room, with 99 TS (24.21%); Category 5, 'Safety culture maturity', with 99 TS (17.85%); and Category 6, 'Fair culture, organizational learning and error reporting', with 57 SP (13.94%) (As you may observe in Fig 1).

We demonstrated the discourses of each category in Table 3.

## Data integration

In Table 4, we resumed and presented the results of the integrated data collected from the quantitative and qualitative approach, using the joint display technique, following the PIP model for data integration in mixed methods research.

We used the PIP technique for data integration to achieve a clear presentation of the relationship between quantitative and qualitative approaches. This integration technique approach

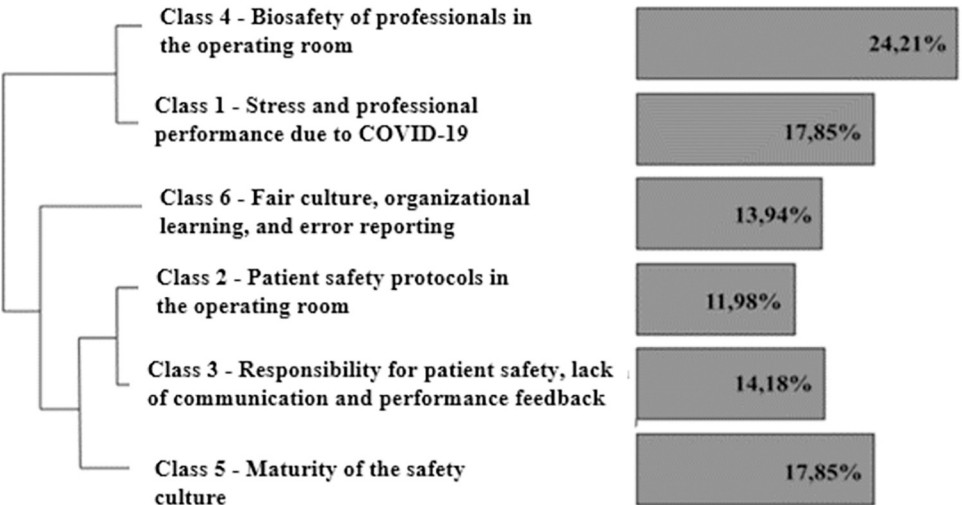

**Fig 1. Descending hierarchical classification Dendrogram.** Note: Survey data (2022) supported by Iramuteq software version 0.7 alpha 2.

made it possible to comparatively assess the impact of COVID-19 on the safety climate in the operating room.

The pillar categories 'Perception of communication in the operating room', 'Perception of stress due to COVID-19' and 'Perception of professional performance due to COVID-19' displays divergences in the quantitative and qualitative approaches. The pillar categories 'Security

**Table 3. Participants' speeches in the qualitative approach.**

| Classes | Discourses |
|---------|-----------|
| Class 1—Stress and professional performance due to COVID-19 | '[. . .] but during the pandemic stress increased.' (Nurse 8) |
| | '[. . .] but during the pandemic, stress has increased.' (Anesthesiologist 17) |
| Class 2—Patient safety protocols in the operating room | '[. . .]But about doctors, I didn't see that much. I saw that they were very dismissive of patient safety protocols [. . .].' (Nursing Technician 4) |
| Class 3—Responsibility for patient safety, lack of communication and performance feedback | 'I realized that all the teams took responsibility for patient safety.' (Support 20) |
| | '[. . .] communication is very fragile.' (Nurse 19) |
| | '[. . .] We don't get any feedback on our performance. Only scolding when we make a mistake.' (Anesthesiologist 16) |
| Class 4—Biosafety of professionals in the operating room | '[. . .] During the pandemic there may have been greater protection for the team with the use of personal protective equipment.' (Anesthesiologist 17) |
| | 'There is a safety concern, both to protect the patient and the professionals. Both before and now during the pandemic.' (Nursing Technician 2) |
| Class 5—Maturity of the safety culture | '[. . .] so it seems that this culture of safety is really moving forward now.' (Nurse 6) |
| | 'Look, the safety culture needs to improve.' (Anesthesiologist 16) |
| Class 6—Fair culture, organizational learning, and error reporting | 'We don't have the freedom to talk about mistakes. It's blaming and stifling. (Nurse 6) |
| | 'I think people are suddenly afraid to admit a mistake, [. . .] precisely because of the guilt' (Nurse 9) |

**Table 4. Integration of quantitative and qualitative data following the model Pillar Integration Process.**

|  | QUANTData mean | QUANT Categories | PILLAR | QUAL Categories | QUAL Codes |
|---|---|---|---|---|---|
| Pre-pandemic | 75.52 | Communication in the Surgical Environment | **Perception of communication in the operating room** | Class 3 14.18% | "[. . .] communication is very fragile, very fragile [. . .]" |
| Pandemic | 76.82 | | | | |
| Pre-pandemic | 69.75 | Safety Climate | **Evolution of security culture** | Class 5 17.85% | "[. . .] even after the pandemic, things are implemented, but there's still not a very strong culture." |
| Pandemic | 70.21 | | | | |
| Pre-pandemic | 65.68 | Perception of Management | **The vision of management and implementation of protocols** | Class 2 11.98% Class 3 14.18% | "[. . .] The management has done a good job, but it could be better [. . .] |
| Pandemic | 65.13 | | | | "We don't get any feedback on our performance, unfortunately none at all." |
| Pre-pandemic | 62.44 | Working Conditions | **Fair organizational culture** | Class 6 13.94% | "We don't have the freedom to talk about mistakes. It's blaming and stifling." |
| Pandemic | 63.30 | | | | |
| Pre-pandemic | 54.87 | Perception of Stress | **Perception of stress due to COVID-19** | Class 1 17.85% | "So I was very stressed at the time of the pandemic for fear of the disease [. . .] " |
| Pandemic | 54.17 | | | | |
| Pre-pandemic | 29.48 | Perception of Job Performance | **Perception of professional performance due to COVID-19** | Class 1 17.85% | [. . .] My performance is better now because I'm not afraid, but before I was very anxious, so that made my performance worse [. . .]." |
| Pandemic | 28.66 | | | | |

Class 1 (Stress and professional performance due to COVID-19); Class 2 (Patient safety protocols in the operating room); Class 3 (Responsibility for patient safety, lack of effective communication and performance feedback); Class 5 (Safety culture maturity); Class 6 (Just culture, organizational learning and error reporting).

culture maturity', 'The vision of management and implementation of protocols' and 'Fair organizational culture' reveal convergence in the quantitative and qualitative approaches.

It is important to highlight the discrepancy in the PIP. Category 4—Biosafety of the professional staff in the operating room—was not integrated with the results of the quantitative approach using the PIP. We designed the interview script for the qualitative approach based on the domains of the SAQ/CC, however this instrument was created in a typical health situation, and therefore, Category 4, outstands due to the pandemic context of COVID-19 in which biosafety measures may have been more valued.

## Discussion

### Quantitative and qualitative approach

The quantitative approach analyzed the domains 'Safety Climate', 'Perception of Management', 'Working Conditions', 'Perception of Stress', 'Communication in the Surgical Environment' and 'Perception of Job Performance'. We emphasize that the domain 'Communication in the Surgical Environment' was the only one with a positive average and 'Perception of Job Performance' was rated with an average of less than 50. In addition, the domains 'Working Conditions' and 'Perception of Stress' displayed a statistically significant association demonstrating the impact of COVID-19 on the safety climate in the operating room.

The qualitative approach permitted an in-depth analysis of the participants' perception of the phenomenon studied, resulting in six categories: 'Stress and professional performance due to COVID-19', 'Patient safety protocols in the operating room', 'Responsibility for patient safety, lack of effective communication and feedback on performance', 'Biosafety of the professional staff in the operating room, 'Safety culture maturity', and 'Fair culture, organizational learning and error reporting'.

Regarding the characterization of the sample, the predominant category that participated in the present study was nursing, hence corroborating other studies concerning the safety climate in the operating room and reflecting the profile of Brazilian nursing, a profession that is mostly performed by nursing technicians [29–33].

Care provided in the operating room is complex. It encompasses available technology, work processes, patient vulnerability and multidisciplinary action, including nursing staff, medical staff (surgeons and anesthesiologists), radiology and laboratory technicians, administrative staff, among others, all interacting in the same environment, with a technical division of labor exercising their function with autonomy and in line with ethical-legal and technical-scientific conduct [34].

In this team, nursing, stands out. In the Brazilian scenario it is considered the largest health workforce, and it is fundamental when facing the challenges imposed by the COVID-19 pandemic since it is able to move the organizational safety culture towards a positive safety climate [5, 35]. We emphasize that the nursing team is an active component and driver of improvements in perioperative patient safety [13]. In the surgical environment, the nursing professional is recognized as the main coordinator and responsible for all the operative phases [36].

Nursing is an extremely important professional health category, as it is seen as the profession that performs the most actions to improve patient safety in the surgical environment, they are responsible for managing the care of the surgical environment, in addition, this category stood out on the front line of the pandemic [37, 38].

The COVID-19 pandemic has imposed a new routine on health institutions, requiring investment in resources, hiring more professionals, mobilizing health teams and services, reducing the supply of services such as elective surgeries, so that care for COVID-19 patients can be guaranteed [39, 40].

These actions could have a positive impact on the study scenario, however, when comparing the pre-pandemic period with the pandemic period, only the 'Communication in the Surgical Environment' domain showed positive responses, i.e. an average of more than 75 in both periods.

We highlight that this domain is the distinguishing feature of the SAQ/OR, making it specific to the surgical environment, as it refers to the information shared between healthcare professionals that covers material resources, equipment, the surgical team, and the patient's clinical conditions to ensure a safe surgical environment for the patient and professionals [7].

Effective communication fosters the maturing of an organization´s safety culture, strengthens interpersonal relationships between teams, patients, and families, and contributes to the reliability of preventive measures, as it provides the team with an environment that favors changes in the work process with the aim of reducing harm to patients and staff [41, 42].

Research that approaches the safety climate in the operating room using the SAQ/OR instrument also identified the communication domain in the surgical environment as the only domain with a positive perception. Communication failures are the main cause of errors and inconveniences at work and are recognized by the World Health Organization as the second international goal for patient safety, since communication is the basis for quality and safety in health care [13, 15].

We need an action plan to ensure efficient communication between professionals on a horizontal basis. The Global Patient Safety Action Plan 2021–2030 includes widely developed strategies that can be adapted to the context, their implementation will help ensure that patient safety is improved at both national and global levels. Improving communication between professionals is crucial and would prevent millions of adverse events. Communication is the basis of quality and safety in care [43].

It is important to highlight the 'Perception of Job Performance' domain, which had an average score of less than 50. This domain involves the professional's ability to recognize and understand that fatigue and work overload have repercussions on professional performance and, consequently, on the safety of surgical patients [7]. Studies carried out in a non-pandemic context revealed that the professional performance domain had the lowest score when compared to the other domains [15, 30].

The present study discloses that, the result of the professional performance domain needs intervention because it reflects how fragile the safety climate is in health institutions, as professionals do not recognize factors that influence their actions and there is no feedback on their performance, and as a result these professionals may not be satisfied with their work, which is closely related to their performance [13].

The ability of professionals themselves to recognize the factors that intervene in their actions, as well as feedback from leaders, favors quality in patient safety and supports the maturity of the organization's safety culture [44]. However, a multicenter study carried out during the COVID-19 pandemic analyzed the safety climate with a sample of 681 professionals from 08 surgical centers in the southern region of Brazil. For the professional performance domain, all hospitals had a score equal to or greater than 75 points, reflecting a positive safety climate in this domain, even in a pandemic context [3].

In the present study, this negative diagnosis of the perception of performance indicates the need to create and implement improvement actions aimed at professional qualification and improvement as a strategy for maturing the safety climate in the operating room, as this domain, in addition to being a factor of interest for professional satisfaction, reflects the professional and behavioral conduct of the worker and the organization's managers [3, 30].

As for the 'Working Conditions' and 'Perception of Stress' domains, these showed a statistically significant association when relating the pre-pandemic and pandemic periods, demonstrating that the pandemic had an impact on the safety climate in the operating room in the study scenario.

The Working Conditions domain reflects the perception of the quality of the working environment in relation to patient safety. This determines that, regardless of the adversities and moments of crisis that may arise, investing in work environments can have positive repercussions. In this research, although the mean score for the domain improved during the pandemic period, it remained below 75, which we consider a negative perception, as well as in other studies [15, 40, 45].

There is evidence linking negative working conditions with the intention to quit the profession [46]. This critical result highlights the need for institutional support, i.e. from senior management in terms of patient safety, enabling conditions and actions that favor safe care. The discourses, resulting from the qualitative approach, reveal that managers were concerned with providing means to guarantee the safety of its professionals during the COVID-19 pandemic.

Potential strategies for improving the working conditions of health professionals in the operating room include the availability of material resources, personal protective equipment, training, adequate working hours and human resources, as it is believed that health professionals are predisposed to making mistakes due to excessive workload and shortage of resources [47, 48].

Human resources are therefore an important component in developing the capacity to promote a positive safety climate and favorable working conditions. In the United States the ratio of nurses per 1000 inhabitants is 12, while in China it is 1000:3.3, in Portugal the ratio is 1000:7.3 nurses and in Italy the ratio is 1,000:6.6 nurses [5, 49, 50].

In Brazil, according to data from the Organization for Economic Co-operation and Development (OECD), the ratio per 1,000 inhabitants is 1.6 nurses. Data collected by the Lancet

Commission on global surgery recorded a surgical workforce ratio of 55 professionals per 100,000 inhabitants in 2018 [51].

In fact, the COVID-19 pandemic has established a new routine in institutions, thus it necessary to create strategies that minimize the possibility of incidents resulting from the high workload, shortage of human and material resources [38]. In the face of the pandemic, health institutions have had to offer constant updates to improve their knowledge of COVID-19 and thus provide appropriate care [52].

A safe working environment includes improving working conditions, redefining care packages, and adopting protocols based on COVID-19 recommendations [53]. These results are indicators for the operating room manager and can be used as a guide for planning and implementing actions with the aim of providing a safe working environment [15].

Reorganization of the care provided in the operating room was necessary to maintain the safety of patients and workers. To this end, practices have been adopted to prevent and control infections caused by COVID-19 in surgical procedures, such as: reducing the number of professionals in the anesthetic-surgical act as much as possible, changing operating room professionals only in cases of emergencies, careful assessment of the risks and benefits of performing the surgical procedure [54].

As for the Perception of Stress, this domain addresses the professionals' recognition of the extent to which stressful factors affect their work. Undoubtedly, the pandemic has influenced the activities carried out by the professionals. The lack of knowledge of the behavior of the disease, the number of deaths among the population and health professionals, has had an emotional impact and, consequently, it has affected the performance of the health teams [4, 5, 55].

The analysis of this domain's statements refer that the professionals surveyed recognize that their performance is impaired in situations such as excessive workload, tense situations, among others. However, in this study, as in other studies, professionals did not recognize external factors as possible influencers in the occurrence of incidents [5, 33, 49, 56, 57].

This dichotomy between the results may be due to the fact that, when answering the SAQ/OR, participants consider situations inherent to work activities to be stressful factors. By exploring the depth of the problem studied using a qualitative approach, its implications become clear.

The discourses also reveal that, for some participants, the perception of stress did not change from the pre-pandemic period to the pandemic period. This was due to the reduction in surgeries, especially during the first wave of COVID-19, in addition to the perception of an improvement in their work performance and in the development of more empathy for patients and their families and valuing safety measures. Empathy functions as a psychological comfort that helps promote social and professional behavior and helps control stress levels [58].

On the other hand, the statements also confirm that the professionals perceived an increase in stress, as in other studies [13, 54, 58] and, consequently, a reduction in the perception of professional performance due to the fear of an unknown disease, fear of becoming infected or infecting their family members, as well as work overload due to the reduction in the team and emotional pressure as they are on the front line fighting the disease.

The COVID-19 pandemic highlighted the enormous challenges and risks faced by health workers worldwide. In 2020, the WHO's campaign "Health worker safety: a priority for patient safety" aimed to focus the attention of patients, health professionals and governments to work together globally to ensure the safety of patients and health workers in every nation on the planet. In this context, among the 16 key challenges identified by WHO, Challenge 1 is to create a culture of safety: a safe and trustworthy environment where the basis of transparency, safety, trust and accountability is established and maintained between the staff of the institution and the patients it serves [59].

In fact, COVID-19 caused changes in the surgical environment. Recognizing stress factors and professional performance contributes to actions that improve health quality and strengthen the safety culture. Management support in recognizing these factors and taking action to provide a good working environment are predictors of safety climate [33].

It is essential that professionals recognize the stressors at work as possible influencers of the quality of care, so that they feel comfortable assuming their physical or mental condition of stress, without fear of reprimand, guaranteeing harm-free care. Work-related stress is indeed an imperative element that negatively influences job security and efficiency. The operating room is characterized by high pressure and a high workload, and this tense working environment has a negative impact on professionals, which can manifest itself in increased infection rates or patient mortality [60–63].

A healthy working environment, which does not neglect the negative impact of stressors, favors the reduction of pressure and fatigue at work, contributing to high-quality care [62, 64]. Given this evidence, gaining this understanding favors an assertive approach by leaders to meet the needs of the patient, their family and surgical team to strengthen the safety climate in the operating room.

## Data integration

Integrating data made it possible to identify convergences, divergences, and discrepancies in the results of the quantitative and qualitative approaches. We defined six pillars: Perception of communication in the operating room; Evolution of safety culture; Vision of management and implementation of protocols; Fair organizational culture; Perception of stress due to COVID-19 and Perception of professional performance due to COVID-19.

There is divergence in the professionals' perceptions when comparing the quantitative category 'Communication in the operating room' to Category 3 'Responsibility for patient safety, lack of effective communication and performance feedback'. The domain that analyzes communication in the surgical environment was the only one with a positive evaluation during the pre-pandemic period, with an average of 75.52, and in the pandemic period, with an average of 76.82.

However, considering the qualitative approach, which sought to understand the professionals' perception of this domain, in both assessment moments, the statements express that the participants consider that communication could be more open between peers, between multi-professional teams and between leaders.

Communication is a non-technical skill that all professionals and organizations should improve since failures in the exchange of information essential to the quality of care provided can lead to stress and frustration among the team [65]. Amongst the most critical weaknesses of the security climate, we find open communication, the exchange and transfer of information and the non-punitive response to errors [66].

This domain is what differentiates SAQ/CC from SAQ, demonstrating its importance for the surgical environment. It is possible to strengthen the communication skills of both teams and managers, as well as the domains of the safety climate, through training to improve care practice in the operating room. The more information the team acquires, the more the organization matures, and, with the support of management, it can significantly strengthen aspects related to patient safety [67].

Communication in the healthcare environment is extremely important, especially in the surgical context. It should be featured that communication is one of the ten essential objectives of the global challenge of the "Safe Surgery Saves Lives Manual" [68]. Despite its importance, effective communication in the surgical environment is still complex and challenging, mainly

due to the differences in values, habits, beliefs, understandings, and experiences of professionals working as a team, as observed in a study carried out in Sweden involving nurses, surgeons, and anesthesiologists [69].

There is evidence that some tools can help with communication in the surgical environment, such as the safe surgery checklist, which minimizes the occurrence of harm to patients and communication failures, and the use of WhatsApp as a technological tool that helps collect, process, store and exchange information in healthcare environments and also contributes to reducing the hierarchy between the professionals of the surgical team [69–71].

The second pillar—Evolution of safety culture—identified convergence of results. The 'Safety climate' domain showed a negative score in both periods, which was confirmed by the discourses in Category 5 'Safety culture maturity'. This domain reflects the professionals' perception of the organization's commitment to patient safety. The implementation of improvement through actions plans is essential to strengthen the safety climate in the operating room, since risky situations surround professional practice, requiring organizations to implement strategies that develop a climate of safety, reflecting safe care with minimized risks [30].

Studies indicate a negative perception in the 'Safety climate' domain, one in the southern region of Brazil with an average of 69.34; and in the central-western region with an average of 65.17. In Turkey, the average was 15.25 [47, 72, 73]. Safety climate must be analyzed in conjunction with the characteristics of the organization and how professionals perceive themselves in the organization, contributing to the maturity of the organization's safety culture.

As a means of strengthening the safety climate in the surgical center, we propose that the unit's managers should form study groups and provide training for the professionals to develop their technical and scientific knowledge, in addition create a space in which the professionals can express their doubts and propose improvements for the unit. Hence, these professionals will perceive themselves as fundamental members within the organization and understand that more information and trust, means that it is possible to achieve a more mature organizational and professional culture.

In 1997, a culture maturity model was developed by Westrum, and adapted by Hudson in 2003, in which there is a range of cultures that mature as the level of information and trust increases. The evolution of security culture maturity is described in five levels. The first phase is, pathological, in which safety is a problem caused by workers and not openly managed; at the reactive level, the organization begins to consider safety as something important after an incident has occurred; at the calculative level, the safety management system is boosted by the increase in information and experiences; at the proactive level, the organization's professionals begin to take safety initiatives with a view to improving processes; when the last level is reached—generative—safety is perceived as a fundamental organizational investment [74].

Professionals who make up the surgical environment are only able to mature culturally, if their leaders and senior management take an assertive look at the professionals' perception of the safety climate in relation to the needs of the care team, providing a maturing environment, with the aim of evolving to the highest level of safety culture, associating the fundamental standards of an organization with values, attitudes, skills, behavior and commitment to health and safety management [45, 74].

Professionals report in their statements that the surgical center under study is moving towards the generative culture maturity level, as there was a concern on behalf of management to discuss patient and professional safety issues in the study setting, establish protocols and provide adequate personal protective equipment in the pandemic context.

The third pillar is the managers vision and the implementation of protocols, which derives from the integration of the quantitative category 'Perception of management' and categories 2

and 3, 'Patient safety protocols in the operating room' and 'Responsibility for patient safety, lack of effective communication and performance feedback', respectively.

Categories 2 and 3 belong to the same thematic axis and reflect the professionals' perception of management during the pre-pandemic and pandemic periods of COVID-19. In both approaches, there is a common understanding that management has acted positively considering the atypical context in which we find the study scenario, and it is crucial to reflect on which actions we need to take to ensure that this domain is no longer fragile.

The domain 'management perception' relates management actions in favor of safety. This domain reflects the perception that professionals have of the hospital, of the local administration, of appropriate feedback on professional performance, and of job satisfaction [7]. The study that compared the perception of the medical team and the nursing team of the safety climate in the operating room, for the domain 'management perception', revealed that both had a negative perception, with the average being 58.24 and 66.19, respectively [15].

In a global overview, a study that compiled research on professionals' attitudes towards patient safety carried out in Europe (Poland, Norway, Sweden, and Albania), Asia (China, Turkey, Saudi Arabia, and Iran), Africa (Kenya), the Americas (Brazil and the United States) and Australia, identified negative management perceptions [49].

On the other hand, we found a study with positive results in the domain 'Management perception'. It also indicated that the study scenario, evaluated by these authors, develops a Safety culture maturity, because there is a close connection between management and initiatives that strengthen safe care [57].

With the intention to strengthen the safety climate, it is important that managers act to build and support a safety culture. This is possible by giving feedback on professional performance and strategies for improving assistance, by distributing resources and by establishing an optimal dimensioning of the human resource structure that enhance safe healthcare and the implementation of actions that mitigate patient harm without penalizing professionals. Management involvement is a key element for disseminating a positive safety climate [10, 41].

Fair organizational culture represents the fourth pillar. The qualitative (Category 6) and quantitative (Working condition) results identify this field as fragile, hence the converging of results. This integration can become an important quality indicator, as a good work environment favors quality assistance, promotes learning from errors, encourages incident notifications, provides means of safe action and acts based on a fair culture.

A study carried out during the COVID-19 pandemic with 103 professionals from the multiprofessional team identified the non-punitive response to errors as a weak field and observed a predominance of underreporting of adverse events [75]. However, a qualitive study portrays the presence of a fair culture in which professionals reveal in their statements that the unit in which they work receives notifications of damages and implements improvement actions based on the understanding of what caused that adverse event [76].

A fair and mature safety culture is reflected in the actions that professionals will employ when working as a form of barrier to the occurrence of incidents resulting from care [77, 78].

A fair safety culture does not look for culprits, above all, it seeks a culture based on justice, the institution recognizes that incidents are due, in many cases, to the organizational system in which the professional is inserted, replacing blame with adequate accountability.

A positive safety culture prioritizes safety above financial and operational goals, encourages and rewards the identification, reporting, and resolution of safety-related issues from the occurrence of incidents, and promotes organizational learning. Therefore, reducing the punitive response to mistakes, encouraging them to report incidents for the purpose of learning from failures, and supportive interaction among colleagues and in the institution can be useful strategies [55].

The fifth pillar is 'Perceived stresses' due to COVID-19. This pillar arises from the comparison of the quantitative category 'Stress perception' with Category 1 'Stress and professional performance due to COVID-19'. Although the quantitative approach demonstrates that professionals do not consider that stressful factors, such as the pandemic, may affect care, the understanding obtained by the qualitative study reveals that professionals understand that stress may have affected patient care, especially in what concerns fear of the unknown. Having knowledge allows managers to implement stress reduction strategies, just as the professional himself can create mechanisms to strengthen this domain [64].

Knowledge of these conditions helps health organizations identify and develop actions to promote and support health professionals [79]. The pandemic scenario has increased stressors in the work environment, which influences the quality of life of professionals and, consequently, can affect the health system because of absenteeism and sick leave [5].

Practitioners can create coping strategies that best suit their lifestyle to decrease stress and promote mental health, such as reading books on mental health, self-help messages, psychotherapy, and meditation [5, 80]. In addition, organizations, in view of these data, can think of strategies such as reduction of working hours, professional appreciation, social support committees at work [81].

The last pillar is 'Perception of professional performance' due to COVID-19. In Category 1, we observed that professionals identified that COVID-19 changed their patient care, as fear triggered other emotional illnesses, which affected their performance in the surgical center. However, this result differs from the quantitative one, in which professionals do not recognize that external factors can affect their professional performance.

The domain 'Perception of professional performance' involves the professional's ability to recognize and understand that fatigue and work overload have an impact on professional performance and, consequently, on the safety of surgical patients, as well as feedback on their performance from the managers [7, 33].

In this way, the negative perception of management favors non-recognition of the factors that affect professional performance. Management committed to safety, favors, and encourages self-knowledge among professionals, providing its staff with care focused on patient safety [13, 15, 31].

Normally, public health emergencies, such as outbreaks and pandemics, expose health professionals to contamination through direct contact with sick people. Working on the front line is conducive to increased physical and emotional fatigue and consequently affects professional performance. Recognizing these organizational factors is the first step towards safe care [4, 5, 64].

It is important to note that Category 4 –'Biosafety of the professional staff in the operating room'—was not interrelated with the quantitative results using the PIP. The interview script for the qualitative approach was drawn up based on the domains of the SAQ/OR, but this instrument was created in a typical health situation, thus Category 4 is highlighted due to the pandemic context of COVID-19 in which biosafety measures may have been more valued.

In view of the findings and the relationship found in the impact of the pandemic on the safety climate in the surgical environment, we suggest that improvement strategies be implemented, such as holding conversation circles where professionals can share their doubts, emotions, and suggestions.

In this manner, management could use these groups to demonstrate the importance of professionals recognizing the factors that affect work activities and, consequently, work safety, in an environment in which there is no judgment.

In addition, conducting these groups can strengthen communication in the surgical environment.

We also suggest a roadmap for the elaboration of an improvement strategy based on the methodology recommended by the Institute of Healthcare Improvement (IHI), in which the strategies can be delineated by the participation of the professionals working in the sector, hence their participation can stimulate a more proactive communication and professional appreciation. We propose to place three key questions for each domain, namely: What are we trying to accomplish? How will we know if this change is an improvement? What changes can we make to improve? Proposals must then be submitted to the PDCA (plan, do, check, act) cycle, also proposed by the IHI [82].

In phase P, we added the 5W2H tool, which enables the elaborator to have a broader view of the actions of a plan. By utilizing this tool, we obtained answers to the following questions: Where? When? Who? What? Why? How? How much? In phase D, the proposed interventions will be presented to the service´s board of directors for approval and to the team, so that they can be implemented in the routine, according to the plan,

We propose using the SAQ/OR instrument to carry out phases C and A, and it should be applied once more to assess whether there was a significant change in the safety climate of the operating room after the implementation of the improvement proposal.

The impact of COVID-19 on the domains of the safety climate highlights the need to invest in the diagnosed weaknesses so that emergencies do not negatively influence the safety culture in the operating room.

## Limitations

Despite the importance of its findings, the present study has limitations, including the fact that it was restricted to the operating room of a single university hospital, and it is not possible to generalize its results to other health institutions. Another limiting factor is that the study was carried out during the COVID-19 pandemic period, making a comparison with the pre-pandemic period simultaneously.

For this reason, and because the research involved self-reporting, systematic failures can occur, as participants may not remember previous events or experiences accurately or omit details. In this way, the accuracy and information of the data from memory can be influenced by subsequent events and experiences. Therefore, the risk of memory bias must be considered.

## Conclusion

This study analyzed the safety climate in the operating room comparing the pre-pandemic and pandemic periods of COVID-19, hence achieving the proposed objective. COVID-19 has had an impact on the safety climate in the operating room, and it associates the pandemic with the 'Perception of stress' and 'Working conditions' domains.

The qualitative analysis of the data allowed us to deepen our knowledge of the phenomenon under research, enabling us to understand how healthcare professionals perceive patient safety. From a qualitative point of view, the pandemic has had an impact on professional performance, and on the perception of stress and communication in the surgical environment; however it has strengthened the recognition of biosafety, reaffirming that there is no patient safety if the professionals are not also safe.

Considering the complexity of the object of the study, the mixed method allowed an understanding of the inferences through the integration of data, enabling an in-depth understanding of the research problem and a more robust assessment of the safety climate in the operating room during the pre-pandemic and pandemic period of COVID-19.

We suggest future comparative, multicenter studies using the SAQ/OR instrument to identify the impact of the COVID-19 pandemic on the safety climate in the operating room, with proposals for improvement strategies in the domains with potential for strengthening.

We therefore hope that this research can contribute to practice, as it identifies the weaknesses of the environment, enabling the implementation of assertive strategies for safe care; to managers, as it provides support for the development of educational actions favoring a positive safety climate; to society, as by diagnosing the weak points, actions are taken to ensure that practice is based on evidence, creating a safe and high-quality health care environment.

## Author Contributions

**Conceptualization:** Rosilene Alves Ferreira, Cintia Silva Fassarella, Olga Maria Pimenta Lopes Ribeiro, Rosane Barreto Cardoso, Danielle de Mendonça Henrique, Flávia Giron Camerini, Rogério Marques de Souza, Ricardo de Oliveira Meneses.

**Formal analysis:** Rosilene Alves Ferreira, Cintia Silva Fassarella, Olga Maria Pimenta Lopes Ribeiro, Danielle de Mendonça Henrique, Flávia Giron Camerini, Rogério Marques de Souza, Ricardo de Oliveira Meneses.

**Investigation:** Rosilene Alves Ferreira, Cintia Silva Fassarella.

**Methodology:** Rosilene Alves Ferreira, Cintia Silva Fassarella, Olga Maria Pimenta Lopes Ribeiro, Rosane Barreto Cardoso, Danielle de Mendonça Henrique, Flávia Giron Camerini, Rogério Marques de Souza, Ricardo de Oliveira Meneses.

**Project administration:** Rosilene Alves Ferreira, Cintia Silva Fassarella.

**Supervision:** Rosilene Alves Ferreira, Cintia Silva Fassarella.

**Validation:** Rosilene Alves Ferreira, Rosane Barreto Cardoso, Flávia Giron Camerini.

**Writing – original draft:** Rosilene Alves Ferreira, Cintia Silva Fassarella, Olga Maria Pimenta Lopes Ribeiro, Rosane Barreto Cardoso.

**Writing – review & editing:** Rosilene Alves Ferreira, Cintia Silva Fassarella, Olga Maria Pimenta Lopes Ribeiro, Rosane Barreto Cardoso, Danielle de Mendonça Henrique, Flávia Giron Camerini, Rogério Marques de Souza, Ricardo de Oliveira Meneses.

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
