## [Editor Report · Decision Letter 0]

4 Feb 2024

PONE-D-23-43150Safety climate in the operating room in the pre-pandemic and pandemic period of COVID-19: A mixed method studyPLOS ONE

Dear Dr. Ferreira,

Thank you for submitting your manuscript to PLOS ONE. After careful consideration, we feel that it has merit but does not fully meet PLOS ONE’s publication criteria as it currently stands. Therefore, we invite you to submit a revised version of the manuscript that addresses the points raised during the review process.

We look forward to receiving your revised manuscript.

Kind regards,

Ebtsam Aly Omer Abou Hashish, 

Academic Editor

PLOS ONE

Additional Editor Comments:

Academic Editor's Comments:

Safety Climate in the Operating Room during the Pre-Pandemic and Pandemic Periods of COVID-19: A Mixed Method Study

We appreciate the authors' significant efforts in addressing an important topic. After carefully reviewing the manuscript, we kindly invite the authors to consider the following points for further refinement before resubmitting the paper:

Abstract:

1. Consider rephrasing the purpose of the study for clarity, such as specifying the focus on safety climate.

2. The convergence strategy is mentioned but not elaborated. Include a brief explanation of how the mixed-method approach was applied.

3. Specify the duration of the pre-pandemic and pandemic periods.

4. Mention the institutional review board approval for ethical considerations.

5. Provide more context regarding the Pillar Integration Process (PIP) for better understanding.

Introduction:

1. Clarify the relevance of analyzing safety climate in the context of COVID-19.

2. Elaborate on the impact of COVID-19 on the safety climate.

3. Improve flow and coherence in the introduction section for better readability.

Aim:

1. The research problem is stated, but it could be more succinct. Consider refining the problem statement.

2. Clarify the Pillar Integration Process and its role in achieving the objectives.

3. Consider rephrasing objectives for better clarity and alignment with the study's focus.

Methods:

1. Provide a concise rationale for choosing mixed-method research in the context of public health emergencies.

2. Specify the role of the Pillar Integration Process in data analysis.

3. Clarify why the specific reporting guidelines and tools (STROBE®, COREQ, MMAT) were chosen.

4. Highlight the importance of intentional sampling in the qualitative phase.

5. Consider restructuring the section for improved organization and clarity.

Overall:

1. Ensure consistency in terminology and formatting throughout the document.

2. Consider adding more details about the characteristics of the study setting for better context.

3. Address potential redundancies and streamline sentences for better readability.

4. Double-check the order and sequence of the content for logical flow.

Results Comments:

Quantitative Data:

1. Clarify the rationale for choosing the 'Communication in the surgical environment' domain for discussion in the quantitative phase.

2. Ensure that the statistically significant associations in the 'Perception of stress' and 'Working conditions' domains are well-explained and discussed.

3. Clearly present the key findings and their implications for the safety climate during the pre-pandemic and pandemic periods.

Qualitative Data:

• Consider adding a brief summary or introduction to the qualitative findings to provide context before presenting the categories.

• Provide a concise overview of the six categories and their respective proportions to facilitate readers' understanding and Discuss the significance of each category, emphasizing their impact on the safety climate in the operating room.

Data Integration:

• Clarify how the Pillar Integration Process (PIP) was employed for data integration, emphasizing its role in synthesizing the quantitative and qualitative findings.

• Clearly articulate the benefits and insights gained from integrating both types of data, emphasizing any convergences or divergences observed.

Discussion :need organization and elaborations in some points:

• Begin the discussion by briefly reintroducing the six identified pillars or categories to provide a clear framework for the discussion.

• Further elaborate on the significance of nursing as the predominant category, especially in the context of the Brazilian healthcare system. Discuss how this dominance might influence the safety climate.

• Explore in more detail the specific challenges faced by healthcare professionals during the COVID-19 pandemic. Discuss the implications of the pandemic on resources, staff management, and overall patient care.

• Provide a more in-depth discussion on the factors contributing to the negative perception in the 'Working Conditions' domain during the pandemic. Explore potential strategies to improve working conditions for healthcare professionals.

• Discuss the dichotomy between quantitative and qualitative findings regarding stress perception. Explore the reasons for this disparity and its implications for healthcare professionals' well-being and patient safety.

• explain the qualitative findings related to communication, addressing the identified need for improvement. Propose potential interventions or strategies to enhance communication within the surgical team.

• Discuss the negative scores in the 'Safety Climate' domain and how these align with the qualitative findings in 'Safety Culture Maturity.' Propose recommendations for strengthening the safety culture in the operating room.

• Provide a nuanced discussion on the varying perceptions of management. Explore how management actions during the pandemic influenced safety culture and propose strategies for improving managerial support.

• Elaborate on the converging results regarding the fragility of the 'Fair Organizational Culture' pillar. Discuss the importance of a fair culture in promoting patient safety and suggest interventions for improvement.

Limitations:

Emphasize the limitations of the study, such as its single-center focus and potential memory bias. Discuss how these limitations may impact the generalizability of the findings.

Conclusion:

Summarize the key findings concisely and restate their implications for practice, management, and society. Highlight the study's contribution to understanding safety climate during the pandemic and suggest avenues for future research.

---

## [Author Response · Author response to Decision Letter 0]

29 Mar 2024

We would like to thank you for your careful review of the material submitted, which helped to improve the submission PONE-D-23-43150 “Safety climate in the operating room in the pre-pandemic and pandemic period of COVID-19: A mixed method study”.

The adjustments made are listed in rebutall letter.

---

## [Decision Letter · Decision Letter 1]

28 May 2024

Clima de segurança no centro cirúrgico no período pré-pandêmico e pandêmico de COVID-19: estudo de método misto

PONE-D-23-43150R1

Dear Authors

We’re pleased to inform you that your manuscript has been judged scientifically suitable for publication and will be formally accepted for publication once it meets all outstanding technical requirements.

Kind regards,

Ebtsam Aly Omer Abou Hashish

Academic Editor

PLOS ONE

Additional Editor Comments (optional):

-

Reviewers' comments:

Reviewer's Responses to Questions

**Comments to the Author**

1. If the authors have adequately addressed your comments raised in a previous round of review and you feel that this manuscript is now acceptable for publication, you may indicate that here to bypass the “Comments to the Author” section, enter your conflict of interest statement in the “Confidential to Editor” section, and submit your "Accept" recommendation.

Reviewer #1: All comments have been addressed

2. Is the manuscript technically sound, and do the data support the conclusions?

Reviewer #1: (No Response)

3. Has the statistical analysis been performed appropriately and rigorously? 

Reviewer #1: Yes

4. Have the authors made all data underlying the findings in their manuscript fully available?

Reviewer #1: No

5. Is the manuscript presented in an intelligible fashion and written in standard English?

Reviewer #1: Yes

6. Review Comments to the Author

Reviewer #1: no comments of the article "Clima de segurança no centro cirúrgico no período pré-pandêmico e pandêmico de COVID-19: estudo de método misto"

7. PLOS authors have the option to publish the peer review history of their article (what does this mean?). If published, this will include your full peer review and any attached files.

Reviewer #1: No

---

## [Editor Report · Acceptance letter]

14 Jun 2024

PONE-D-23-43150R1 

PLOS ONE

Dear Dr. Ferreira, 

I'm pleased to inform you that your manuscript has been deemed suitable for publication in PLOS ONE. Congratulations! Your manuscript is now being handed over to our production team.

Kind regards, 

on behalf of

Prof Ebtsam Aly Omer Abou Hashish 

Academic Editor

PLOS ONE